# CD112 Supports Lymphatic Migration of Human Dermal Dendritic Cells

**DOI:** 10.3390/cells13050424

**Published:** 2024-02-28

**Authors:** Neda Haghayegh Jahromi, Anastasia-Olga Gkountidi, Victor Collado-Diaz, Katharina Blatter, Aline Bauer, Lito Zambounis, Jessica Danielly Medina-Sanchez, Erica Russo, Peter Runge, Gaetana Restivo, Epameinondas Gousopoulos, Nicole Lindenblatt, Mitchell P. Levesque, Cornelia Halin

**Affiliations:** 1Institute of Pharmaceutical Sciences, ETH Zurich, Vladimir-Prelog-Weg 1-5/10, 8093 Zurich, Switzerland; neda.haghayegh@gmail.com (N.H.J.); anastasia-olga.gkountidi@pharma.ethz.ch (A.-O.G.); victor.collado@pharma.ethz.ch (V.C.-D.); katharina.blatter@pharma.ethz.ch (K.B.); lito.zambounis@pharma.ethz.ch (L.Z.); jessica.medina@pharma.ethz.ch (J.D.M.-S.); russoerica87@gmail.com (E.R.); peter_runge12@hotmail.com (P.R.); 2Department of Dermatology, University Hospital Zurich, University of Zurich, 8091 Zurich, Switzerland; gaetana.restivo@usz.ch (G.R.); mitchell.levesque@usz.ch (M.P.L.); 3Department of Plastic Surgery and Hand Surgery, University Hospital Zurich, Raemistrasse 100, 8091 Zurich, Switzerland; epameinondas.gousopoulos@usz.ch (E.G.); nicole.lindenblatt@usz.ch (N.L.)

**Keywords:** CD112, nectin-2, dendritic cells, lymphatic endothelial cells, lymphatic migration, human

## Abstract

Dendritic cell (DC) migration from peripheral tissues via afferent lymphatic vessels to draining lymph nodes (dLNs) is important for the organism’s immune regulation and immune protection. Several lymphatic endothelial cell (LEC)-expressed adhesion molecules have thus far been found to support transmigration and movement within the lymphatic vasculature. In this study, we investigated the contribution of CD112, an adhesion molecule that we recently found to be highly expressed in murine LECs, to this process. Performing in vitro assays in the murine system, we found that transmigration of bone marrow-derived dendritic cells (BM-DCs) across or adhesion to murine LEC monolayers was reduced when CD112 was absent on LECs, DCs, or both cell types, suggesting the involvement of homophilic CD112–CD112 interactions. While CD112 was highly expressed in murine dermal LECs, CD112 levels were low in endogenous murine dermal DCs and BM-DCs. This might explain why we observed no defect in the in vivo lymphatic migration of adoptively transferred BM-DCs or endogenous DCs from the skin to dLNs. Compared to murine DCs, human monocyte-derived DCs expressed higher CD112 levels, and their migration across human CD112-expressing LECs was significantly reduced upon CD112 blockade. CD112 expression was also readily detected in endogenous human dermal DCs and LECs by flow cytometry and immunofluorescence. Upon incubating human skin punch biopsies in the presence of CD112-blocking antibodies, DC emigration from the tissue into the culture medium was significantly reduced, indicating impaired lymphatic migration. Overall, our data reveal a contribution of CD112 to human DC migration.

## 1. Introduction

Leukocyte trafficking between peripheral tissues and secondary lymphoid organs is essential for optimal immune protection. Most leukocytes continuously migrate throughout the body using blood vessels and lymphatic vessels for rapid transport [1]. Their trafficking is governed by an interplay of chemokines and adhesion molecules expressed on endothelial cells (ECs) and their corresponding chemokine receptors and adhesion molecules expressed on leukocytes [1,2]. These interactions allow specific leukocyte subsets to extravasate from the bloodstream into tissues through a cascade of events that enables cells to arrest on the endothelium and finally transmigrate [2,3]. While selectins expressed by ECs or on leukocytes are important for tethering and rolling of leukocytes in the vasculature [4], subsequent firm arrest involves the binding of leukocyte-expressed integrins to their ligands, i.e., EC-expressed immunoglobulin superfamily (IgSF) members [5], such as the intercellular adhesion molecule 1 (ICAM-1) or vascular cell adhesion molecule 1 (VCAM-1) [2,3,6]. However, besides these integrin-binding IgSF family members, other members such as junctional adhesion molecules (JAMs) [7] and nectin-2 [8,9,10] have recently been found to contribute to leukocyte trafficking across blood vessels. Nectin-2 belongs to a group of four structurally related molecules (nectin 1–4) and is also known as poliovirus-like receptor 2 (PVRL2) or CD112 [11,12]. CD112 engages in homophilic as well as heterophilic interactions with other nectin-binding partners, such as CD113 [13,14]. CD112 also has immunomodulatory properties as it interacts with DNAX accessory molecule 1 (DNAM-1) and the coinhibitory T-cell immunoreceptor with Ig and ITIM domains (TIGIT) [15]. In vitro, CD112 was found to be expressed at blood vascular endothelial cell–cell junctions [16], and it was also detected in high endothelial venules (HEVs) of human LNs and blood vessels of human skin [14,17]. Similarly, our group recently reported CD112 expression in cultured murine blood endothelial cells (BECs) as well as in the murine vasculature of the skin, LNs and spleen [10]. CD112 was found to support in vitro transmigration of monocytes across blood vascular endothelium [17], and a recent study from our group showed that CD112 expression in BECs contributed to angiogenesis as well as to T-cell homing to the spleen [10].

Besides blood vessels, leukocytes also use lymphatic vessels to rapidly travel between peripheral tissues, LNs and the bloodstream [18,19]. These vessels are lined by a single layer of lymphatic endothelial cells (LECs), which are connected by adhesion molecules that regulate lymphatic integrity and permeability as well as the intravasation of leukocytes [19,20]. Next to their essential role in tissue fluid homeostasis, lymphatic vessels are also important for immune function. One of the main migrating, lymph-borne cell populations is DCs, which play a fundamental role in immune homeostasis, surveillance and induction of protective immunity. DCs capture antigens and process and present them on major histocompatibility complex (MHC) to induce an immune response or to contribute to the maintenance of tolerance [21,22]. The DC-expressed chemokine receptor CCR7 and its lymphatic-expressed chemokine ligand CCL21 have long been established as one of the main factors mediating DC migration through afferent lymphatic vessels [23,24,25]. DC migration is an integrin-independent process in a steady state [26], while upon inflammation, it becomes integrin-dependent when the integrin ligands ICAM-1 and VCAM-1 are upregulated in LECs [27,28,29]. Besides ICAM-1 and VCAM-1, several other LEC-expressed adhesion molecules have been shown to contribute to the lymphatic migration of DCs, such as ALCAM [30], CLEVER1 [31], JAMs [7], or the mannose receptor [32].

CD112 was found to be expressed by human monocyte-derived DCs (moDCs) in vitro and by DCs in human LNs in vivo [17]. Moreover, we detected high expression of CD112 in cultured murine LECs as well as in murine lymphatics in vivo [10], which prompted us to investigate the contribution of CD112 to DC migration in human and murine cell culture systems as well as in human and murine tissues.

## 2. Materials and Methods

### 2.1. Mouse Strains

Wild-type (WT) C57BL/6 mice were purchased from Janvier (Genest-Saint-Isle, France). CD112 knock-out (KO) mice [33] and littermate controls (WT) were bred and housed in specific pathogen-free (SPF) conditions. All animal experiments were performed in mice aged 8 to 12 weeks.

### 2.2. Bone Marrow-Derived Dendritic Cells (BM-DCs)

DCs were generated from the bone marrow (BM) of mice as previously described [34]. Briefly, BM was extracted from the tibia and femurs, and red blood cells were lysed using ACK buffer according to the manufacturer’s instructions (BD Biosciences, Allschwil, Switzerland). About 5 × 10^6^ cells were cultured in bacterial dishes (Greiner Bio-One, Kremsmünster, Austria) in 10 mL of DC medium, which contained RPMI 1640 (Sigma-Aldrich, St. Gallen, Switzerland), 10% FCS, 1 mM sodium pyruvate, penicillin (100 U/mL), 15 mM HEPES, L-glutamine (2 mM), streptomycin (100 μg/mL) (all from Thermo Fisher Scientific, Waltham, MA, USA), 50 μM β-mercaptoethanol (Sigma-Aldrich) and 80 ng/mL GM-CSF. The latter was derived from the supernatant of hypoxanthine-aminopterin-thymidine-sensitive Ag8653 myeloma cells (X63 Ag8.653) that had been transfected with murine GM-CSF cDNA [35]. On day 8 or 9, the floating cell fraction was collected and transferred into tissue-culture-treated dishes (TPP), and the DC medium was supplemented with 0.1 μg/mL LPS (Enzo Life Sciences, Farmingdale, NY, USA). After overnight culture, the floating BM-DCs were harvested for further functional assays and for flow cytometry analysis (FACS) to evaluate the purity and maturation, using the following antibodies: APC Armenian hamster anti-mouse CD11c, BV421 rat anti-mouse MHC-II, APC/Fire750 Armenian hamster anti-mouse CD80, BV605 rat anti-mouse CD86 and corresponding isotype controls (all purchased from BioLegend, San Diego, CA, USA). Nectin-2 expression on matured BM-DCs was analyzed by using AF488 rat anti-mouse Nectin-2/CD112 and corresponding isotype control from R&D Systems, Minneapolis, MN, USA.

### 2.3. Human Monocytes-Derived DC (moDC) Differentiation

Human peripheral blood mononuclear cells (PBMCs) were isolated from buffy coats (purchased from the Blood Donation Center, Zurich, Switzerland) using Ficoll (Ficoll paque plus, Sigma-Aldrich) density gradient centrifugation. Isolated PBMCs were incubated with 1× dilution of red blood cell lysis buffer (BD Biosciences) for 15 min at room temperature (RT). After a PBS washing step, PBMCs were magnetically labeled with human CD14 microbeads (Miltenyi Biotec, Zurich, Switzerland) and loaded onto a MACS LS Column, placed in the magnetic field of a MACS Separator based on manufacturer instructions. Purified CD14^+^ monocytes were differentiated to DCs by cultivation of cells in the presence of 50 ng/mL of recombinant human GM-CSF [36] (PeproTech, London, UK) and 50 ng/mL of recombinant human IL-4 (R&D Systems) in RPMI-1640 medium (Sigma-Aldrich) supplemented with 10% FBS, 1% antibiotic antimycotic solution, 1% L-glutamine and 1.5% HEPES buffer (all from Gibco, Waltham, MA, USA) for 7–8 days. Maturation of moDCs was performed by culturing cells in the presence of 500 ng/mL of LPS for the last day (i.e., days 7 to 8).

### 2.4. Isolation and Culture of Lymph Node LECs (LN-LECs)

Primary LN-LECs were isolated and cultured as previously described [10]. In brief, skin-draining LNs (popliteal, inguinal, axillary, brachial and auricular) were isolated from WT and CD112 KO mice. Digestion was performed in RPMI medium supplemented with 0.25 mg/mL Liberase DH (Roche, Basel, Switzerland) and 200 U/mL DNase I (Sigma-Aldrich) for 1 h at 37 °C. Subsequently, cell suspensions were filtered through 70 µm cell strainers and cultured on cell culture dishes pre-coated with 10 µg/mL collagen type I (PureCol, Advanced BioMatrix, Carlsbad, CA, USA) and 10 µg/mL fibronectin (Millipore, Burlington, MA, USA) in Minimal Essential Medium (MEM)-alpha medium, which was supplemented with 10% FBS and 1× penicillin/streptomycin (all from Gibco). Once the cells reached >80% confluency (typically on days 5–7), the plates were a mixture of lymph node stromal cells (LNSCs), fibroblastic reticular cells (FRCs) and LECs. LNSCs were detached with Accutase, for 4–5 min at 37 °C, washed and purified using CD31^+^ microbeads (Miltenyi Biotech, Bergisch Gladbach, Germany). Isolated LECs were seeded on collagen and fibronectin-coated cell culture dishes and kept up to 6 passages after isolation.

### 2.5. Cell Culture of Conditionally Immortalized Murine Lymphatic Endothelial Cells and Human Dermal Lymphatic Endothelial Cells

Conditionally immortalized murine LECs (imLECs) [27], which express a heat-labile version of the large T antigen, were cultured at 33 °C in medium supplemented with 40% DMEM (low glucose), 40% F12-Ham, 20% FBS (all from Gibco, Waltham, MA, USA), 56 μg/mL heparin (Sigma-Aldrich), 10 μg/mL endothelial cell growth supplement (Sigma-Aldrich), 1% antibiotic antimycotic solution (Fluka, Buchs, Switzerland) and 2 nM L-glutamine (Fluka), as previously described [27,29]. Notably, cell culture dishes precoated with 10 μg/mL collagen (Purecol, Advanced Biomatrix) and 10 μg/mL fibronectin (Millipore) were used. Murine IFNγ (1 U/mL, Peprotech, London, UK) was added to the cultured cells to induce large T-antigen expression [37]. After reaching confluency, the medium was exchanged without adding IFNγ, and the cell culture dish was transferred to 37 °C. Forty-eight hours later, imLECs were washed with PBS and detached with Accutase (Sigma-Aldrich) for flow cytometry analysis.

Human primary dermal LECs (PromoCell-Lot. 439Z007.2, p4-p6) were cultured on collagen-coated cell culture dishes (10µg/mL, PureCol, Advanced BioMatrix) in EBM medium (Lonza, Basel, Switzerland) containing supplements and growth factors provided in the EGM-2 kit (Cat#: CC-4176, Lonza). The cells were incubated at 37 °C until reaching the confluency.

### 2.6. Flow Cytometry

Ear and LN single-cell suspensions were prepared as described [27]. Briefly, ear skin was cut into small pieces and digested with 4 mg/mL collagenase IV (Invitrogen, Waltham, MA, USA) in PBS for 45 min at 37 °C. The mixture was then filtered through a 40 μm cell strainer (BD Biosciences). For subsequent FACS analysis, unspecific FcRγ binding was blocked with rat anti-mouse CD16/32 (10 μg/mL, BioLegend) for 10 min at 4 °C. Then the following antibodies or corresponding isotype controls were added for 30 min at 4 °C: APC/Cy7 rat anti-mouse CD45 (BioLegend), BV421 rat anti-mouse CD31 (BioLegend), APC Syrian hamster anti-mouse Podoplanin (BioLegend), PE/Cy7 or APC Armenian hamster anti-mouse CD11c (BioLegend), BV421 rat anti-mouse MHC class II (BioLegend), Alexa Fluor 488 rat anti-mouse CD112 (clone:829038, R&D system) and Zombie Aqua fixable viability dye (dilution as recommended by the manufacturer, BioLegend). CD112-binding partners’ expression on in vitro matured BM-DCs and in vitro cultured primary LN-LECs was analyzed by using the following antibodies and corresponding isotype controls: PE/Cy7 rat anti-mouse CD226 (BioLegend), BV421 mouse anti-mouse TIGIT (BioLegend) and goat anti-mouse CD113 (R&D system). An Alexa Fluor 488-conjugated secondary antibody (Caltag Laboratories, Little Balmer, UK) was used for the detection of uncoupled CD113 primary antibody. Human LECs and DCs phenotypes (profiles) were analyzed by using the following antibodies and corresponding isotype controls: FITC mouse anti-human CD31 (BD Biosciences), 10 µg/mL Alexa Fluor 647 mouse anti-human Podoplanin (Novus Biologicals, Centennial, CO, USA), PE mouse anti-human CD112 (BioLegend), FITC mouse anti-human CD14 (BioLegend), PE mouse anti-human CD11c (BioLegend), APC mouse anti-human CD86 (BioLegend), FITC mouse anti-human HLA-DR (BioLegend) and Alexa Fluor 647 anti-human CD113 (Santa Cruz Biotechnology, Dallas, TX, USA).

All anti-mouse antibodies were used at 2.5 μg/mL and added in FACS buffer (PBS containing 2% FCS (Thermo Fisher) and 2 mM EDTA (Sigma-Aldrich)). Anti-human antibodies were used with the concentration recommended by the manufacturer. After incubation, cells were washed twice with FACS buffer, and the samples were acquired on a CytoFLEX S Flow Cytometer with CytExpert software 2.5 (Beckman Coulter, Brea, CA, USA) and analyzed using FlowJo software 10.4.1 (TreeStar, Ashland, OR, USA).

### 2.7. In Vitro Transmigration Assay

The transwell inserts (TCS004024, Jet Biofil, Guangzhou, China, 5 µm pore size) were coated with collagen/fibronectin (10 µg/mL) for seeding mouse LN-LECs or collagen (10 µg/mL) for seeding human LECs and incubated at 37 °C for 30 min. Then, 40,000 mouse or human LECs were seeded in each insert. After 48 h, each insert of cells was treated with blocking antibodies or corresponding isotype controls for 30 min at 37 °C. In the case of using mouse LN-LECs isolated from WT or CD112 KO mice, the LECs were treated with 10 µg/mL purified rat anti-mouse CD54 (ICAM-1, clone: YN1/1.7.4, Biolegend) or rat IgG2b (clone: RTK4530, Biolegend). For the transmigration assays, 50,000 WT or CD112 KO BM-DCs were added to the apical side of the transwell inserts, and transmigration was induced by adding 100 ng/mL of CCL21 in medium to the basolateral compartment of the inserts. In the case of human DC transmigration, LECs were treated with 20 µg/mL mouse anti-human CD54 (clone: BBIG-II, R&D Systems), 20 µg/mL mouse anti-human CD112 (R2.525, Santa Cruz Biotechnology, Dallas, TX, USA) or 20 µg/mL normal mouse IgG1 (Santa Cruz) isotype control. About 150,000 human moDCs were added to the apical side of the transwell inserts, and transmigration was induced by an FBS gradient. Specifically, the medium in the apical transwell compartment was supplemented with 10% FBS, and the medium in the basolateral compartment was supplemented with 20% FBS. After 2 h (mouse BM-DCs transmigration) or 17 h (human monocytes-derived DCs transmigration), the assay was stopped and the transmigrated cells were collected from the basolateral compartment of the transwell insert. Transmigrated BM-DCs and human monocytes-derived DCs were stained with Armenian hamster anti-mouse CD11c-APC, rat anti-mouse MHCII-BV421, mouse anti-human CD11c-PE and mouse anti-human CD86-APC, and number of DCs was quantified by flow cytometry using a CytoFLEX S Flow Cytometer (Beckman Coulter, Brea, CA, USA).

### 2.8. In Vitro Adhesion Assay

Isolated mouse LN-LECs were seeded on precoated 96-well plates (Corning Costar^®^, Corning, NY, USA) with collagen/fibronectin (10 µg/mL). LN-LECs were treated with blocking antibodies or corresponding isotype controls for 1 h at 37 °C. Then, 10,000 WT or CD112 KO BM-DCs were added on top of treated LECs and incubated for 45 min at 37 °C. After incubation, non-adherent BM-DCs were removed and mouse LECs were washed twice with PBS. LECs were detached with Accutase, and the cell suspensions were stained with Armenian hamster anti-mouse CD11c-APC and rat anti-mouse MHCII-BV421. The number of BM-DCs was quantified by flow cytometry using a CytoFlex S Flow Cytometer (Beckman Coulter).

### 2.9. TPA-Induced Ear Skin Inflammation

Ears of WT and CD112 KO mice were treated by topical application of 2 μg of TPA (12-O-tetradecanoylphorbol-13-acetate, Sigma-Aldrich) dissolved in 20 μL of acetone. Then, 10 μL of TPA was applied on isoflurane-anesthetized mice, on each side of the ear. After 24 h, ear thickness was measured using a caliper (Brütsch Rüegger, Urdorf, Switzerland), and animals were sacrificed. Single-cell suspensions were generated, according to the flow cytometry protocol, to determine CD112 expression on LECs and BECs or on endogenous DCs present in inflamed ears.

### 2.10. Induction of a Contact Hypersensitivity (CHS) Response in Footpad

A CHS response was induced as previously described [27]. In brief, isoflurane-anesthetized CD112 KO mice and WT littermate control mice were sensitized by topical application of 2% oxazolone (4-Ethoxymethylene-2-phenyl-2-oxazolin-5-one; Sigma-Aldrich) in acetone/olive oil (4:1 volume/volume) on the shaved abdomen (50 µL) and on each paw (5 µL). Five days later, 1% oxazolone was applied to one footpad of each mouse.

### 2.11. FITC Painting Experiment

The ears of mice were inflamed by topical application of TPA. After 24 h, fluorescein-5(6)-isothiocyanate (FITC) (5 mg/mL, Sigma-Aldrich) was dissolved in acetone and dibutyl phthalate (1:1, Sigma-Aldrich), and 20 μL was applied to each side of the ear. After 20 h, ear-draining auricular LNs were harvested and passed through 40 μm cell strainers. Cell suspensions were stained with PE/Cy7 Armenian hamster anti-mouse CD11c and BV421 rat anti-mouse MHC-II (both from BioLegend). For each sample, a total number of migratory DCs (CD11c^+^ MHCII^+^) were acquired on a Cytoflex S, and the percentage and number of FITC^+^ and FITC^−^ migratory cells were analyzed in FlowJo. The number of FITC^+^ migratory cells was normalized to the cellularity of corresponding auricular LNs.

### 2.12. Adoptive Transfer of BM-DCs in the Footpad of Mice

After 24 h CHS challenge, labeled BM-DCs were injected into the inflamed footpads of CD112 KO mice and WT mice, as previously described [29]. For that purpose, LPS-matured WT and CD112 KO BM-DCs were labeled with CMFDA or DeepRed (DR) dyes. After labeling and washing with PBS, 0.5 million labeled WT BM-DCs were mixed with 0.5 million labeled CD112 KO BM-DCs (ratio 1:1) and injected in the total volume of 10 µL into the CHS-inflamed or steady-state footpad of WT and CD112 KO mice (total of 1 million cells in 10 µL sterile PBS per each footpad). After 20 h of adoptive transfer of BM-DCs to the footpad of mice, the popliteal LNs were harvested and passed through 40 μm cell strainers. The single-cell suspensions were stained with BV421 rat anti-mouse MHC-II and PE/Cy7 Armenian hamster anti-mouse CD11c for 30 min at 4 °C. The samples were acquired on a CytoFlex S, and the total number of migratory labeled (CMFDA or DR) BM-DCs were analyzed using FlowJo software version 10.9.0.

### 2.13. Isolation of the Stromal Vascular Fraction (SVF) from Human Skin

Healthy skin of the abdomen and arm tissue from two different donors were obtained from the plastic surgery department of the University Hospital Zurich. Only biopsies from donors who had given written consent were used for further analysis. Skin tissues were first washed with Hank’s Balanced Salt Solution (HBSS) supplemented with 5% FBS, 2% antibiotic–antimycotic solution (AA, Gibco) and 20 mM HEPES (all from Gibco). The skin was minced and digested enzymatically in 1000 U/mL collagenase type I (Worthington, Columbus, OH, USA, LS004197) and 40 µg/mL Dnase I (Roche, 11284932001) in RPMI 1640 GlutaMAX medium supplemented with 10% FBS and 1% AA (all from Gibco) for 1 h at 37 °C under constant agitation. Digested skin tissues were smashed with a plunger and filtered through a 100 µm cell strainer, further washed with supplemented RPMI medium and centrifuged at 300 g for 8 min. The suspension was filtered again through a 100 µm cell strainer. The resulting SVF was cryopreserved in a total of 1 mL of 90% FBS and 10% DMSO (long-term storage in liquid nitrogen).

### 2.14. Flow Cytometry Detection of LECs and DCs from Human Skin SVF

Single frozen vials of the SVF were thawed at 37 °C in a water bath and the content was transferred to 9 mL of RPMI 1640 GlutaMAX medium supplemented with 10% FBS and 1% AA (all from Gibco) before centrifugation at 300 g for 5 min. Cells were incubated first with human Fc receptor blocking solution (1:10, Biolegend, 422302) and efluor 780 fixable viability dye (1:1000, eBioscience, San Diego, CA, USA, 65-0865-14,) in PBS for 5 min at 4 °C. Afterwards, the antibodies were directly added on the cell suspension and incubated for 15 min at 4 °C. For staining of LECs, the following antibodies were used: BV421 anti-human CD45 (1:200, Biolegend, 304032), FITC anti-human CD31 (1:20, BD Pharmingen, NJ, USA, 555445,), Pe-Cyanine 7 anti-human podoplanin (1:200, Biolegend, 337014) and PE anti-human Nectin-2 (1:200, Biolegend, 337409) or the isotype PE anti-mouse IgG1 (1:200, Biolegend, 400114). For staining of DCs, the following antibodies were used: BV421 anti-human CD45 (1:200, Biolegend, 304032), APC anti-human CD86 (1:200, Biolegend, 305411), FITC anti-human HLA-DR (1:200, Biolegend, 307603) and PE anti-human Nectin-2 (1:200, Biolegend, 337409) or the isotype PE anti-mouse IgG1 (1:200, Biolegend, 400114). Cell suspensions were washed once prior to recording on a CytoFlex S instrument (Beckman Coulter). Data were analyzed using FlowJo software version 10.8.1 (BD Life Sciences).

### 2.15. Immunofluorescence Staining of Human Skin

Surplus biopsies from normal human skin were obtained from the biobank of the Department of Dermatology, University Hospital Zurich (EK647-PB_2018-00194), with the assistance of the SKINTEGRITY.CH project. Only samples from patients who had signed an informed consent were used in this study.

Cross-sections of embedded human skin were prepared in a CryoStar NX50 (Thermo Fisher) in 14–16 μm thickness. Sections were dried in RT for 5 min and then fixed in ice-cold acetone (−20 °C) for 2 min at RT and then in 4 °C cold methanol for 5 min in RT. After washing the sections 3× for 6 min in TBS-T at RT, the slides were dried and blocked for 1 h in Immunomix (5% normal donkey serum, 10% BSA, 10% Triton X in PBS). Incubation with primary antibody was performed o/n at 4 °C in a humid and dark chain container. Sections were incubated, the following day, with the secondary antibody for 1 h at RT, followed by wash 2× for 5 min with PBS in a rotator. Slides were washed once more with TBS-T at RT in a rotator followed by mounting with Moviol.

### 2.16. Crawl-Out Assays

For human skin, abdominal or breast punch biopsies of 6 mm diameter were placed in DMEM medium (Invitrogen, Waltham, MA, USA), supplemented with 10% FCS (Thermofisher). From each donor, 4–10 punch biopsies were taken and placed without antibody or with isotype control antibody (Biolegend-401408) or CD112-blocking antibody (clone R2.525) for 48 h at 37 °C. Afterwards, cells that had crawled out into the culture medium were stained and analyzed by flow cytometry.

For mouse ear skin tissue, ears from WT and KO mice were split along the cartilage, and dorsal and ventral sides were placed facing down in DC medium containing RPMI 1640 (Sigma-Aldrich), 10% FCS, 15 mM HEPES, 1 mM sodium pyruvate, penicillin (100 U/mL), streptomycin (100 μg/mL), L-glutamine (2 mM) (all from Thermo Fisher Scientific), 50 μM β-mercaptoethanol (Sigma-Aldrich) and 80 ng/mL GM-CSF (derived from the supernatant of hypoxanthine-aminopterin-thymidine-sensitive Ag8653 myeloma cells (X63 Ag8.653) transfected with murine GM-CSF cDNA3) [35]. After 48 h, crawled-out cells were harvested from the medium and stained for analysis by flow cytometry, and single-cell suspensions of DCs and LECs of mouse ear tissues were obtained by enzymatic digestion with 4 mg/mL collagenase IV (Invitrogen, Basel, Switzerland) for 45 min at 37 °C and subsequently passed through a 40 μm cell strainer (Invitrogen), as described previously [27].

### 2.17. Statistical Analyses

Statistical analysis and graph preparation were performed using GraphPad Prism 10 software (GraphPad Software). Normal distribution was evaluated by applying the Shapiro–Wilk normality test. Depending on the outcome, paired (line-connected dots in graphs) Student’s *t*-test or unpaired Mann–Whitney test was used for comparisons between two groups.

## 3. Results

### 3.1. CD112 Is Expressed in Murine BM-DCs and LECs and Supports DC Transmigration

Previous work from our group has shown that CD112 is expressed in LECs and BECs in murine tissues [10], whereas other studies have reported its expression by human moDCs as well as by endogenous DCs in human LNs [17]. To investigate whether murine DCs also expressed CD112, we performed flow cytometry analysis (FACS) on in vitro-generated bone marrow-derived DCs (BM-DCs) (Figure 1A). Both immature and LPS-matured BM-DCs (CD11c^+^ MHCII^+^) expressed low levels of CD112, which was significantly increased upon LPS-induced DC maturation (Figure 1A,B). As expected, antibody-based detection of CD112 was abolished in LPS-matured BM-DCs generated from CD112-deficient (KO) mice, confirming the specificity of the staining (Figure 1C,D). In comparison to its expression in BM-DCs, flow cytometry using the same fluorescently labeled antibody detected higher CD112 levels in CD31^+^ podoplanin^+^ primary murine LECs isolated from LNs (LN-LECs) (Figure 1E,F). To investigate whether CD112 contributes to DC transmigration, in vitro transmigration experiments were performed using LPS-matured BM-DCs and LECs derived from either WT or CD112 KO mice. To this end, LN-LECs were grown to confluence in transwell inserts, and BM-DCs were added to the apical side of the transwell insert and left to transmigrate towards the medium on the basolateral side of the transwell insert, which had been supplemented with the chemoattractant CCL21 (Figure 1G). In line with previous findings [27,29,38], subsequent FACS-based quantification of transmigrated DCs retrieved from the basolateral compartment of the transwell revealed a strong reduction in the number of transmigrated DCs when the experiments had been performed in the presence of an ICAM-1-blocking antibody (Figure 1H). Similarly, when CD112 was absent on either DCs or LECs, the number of transmigrated cells was significantly reduced, in comparison to the WT controls (Figure 1I,J), as well as when both DCs and LECs did not express CD112 (Figure 1K). We also performed an adhesion assay, using different combinations of WT and KO LECs and WT and KO BM-DCs. Similar to the results of the transmigration assay, DC adhesion to LECs was strongly reduced when CD112 was absent in either DCs or LECs (Figure 1L). Interestingly, no further reduction in adhesion was achieved when CD112 was missing in both LECs and DCs (Figure 1L). Overall, these data suggest that homophilic CD112-mediated DC-LEC interactions supported murine BM-DC adhesion and transmigration. To further investigate this hypothesis, we analyzed the surface expression of other CD112-binding partners, such as DNAM-1 (CD226), TIGIT and CD113 [14,15,39] (Appendix A). Neither the murine BM-DCs nor the LN-derived LECs used in our experiments expressed any of the aforementioned markers, in support of our conclusion that LEC- and DC-expressed CD112 likely engaged in homophilic interactions.

### 3.2. CD112 Is Highly Expressed in Murine Dermal LECs, Whereas Murine Dermal DCs Only Express Low CD112 Levels

Having observed the expression of CD112 in BM-DCs and LECs in vitro (Figure 1) prompted us to further investigate whether CD112 was also expressed in murine DCs and LECs in vivo in murine skin. In agreement with our previous report [10], CD112 expression was detected in LECs and was consistently higher in LECs as compared to BECs (Figure 2A,B). To investigate how tissue inflammation would impact CD112 expression in LECs, phorbol ester 12-O-tetradecanoylphorbol-13-acetate (TPA) was topically applied to the ear skin of WT mice (Figure 2C) [40,41]. The inflamed ear skin and auricular LNs draining the ear skin were analyzed 24 h later. Flow cytometry revealed that CD112 levels in dermal LECs were only slightly increased in four out of five experiments performed (Figure 2D,E). In contrast, while LECs in ear-draining auricular dLNs expressed very low levels of CD112 in a steady state, CD112 expression was consistently upregulated in LN-LECs upon TPA-induced skin inflammation (Figure 2F,G). Different from its expression in dermal LECs, CD112 levels in dermal DCs were extremely low, and no clear upregulation of CD112 was observed during TPA-induced tissue inflammation (Figure 2H,I). To more conclusively assess whether skin-emigrating migratory DCs express CD112, we performed an ear skin crawl-out experiment (Figure 2J). The latter assay relies on the fact that when murine or human skin is excised and taken into the culture, DCs in the tissue start to mature, likely because of the release of danger-associated molecular patterns (DAMPs), migrate towards and into lymphatic vessels [25,42,43,44] and, subsequently, exit the tissue by crawling out through lymphatics into the culture medium, where they can be detected by FACS [30,38]. Performing this assay, we detected a clear difference in the CD112 signal present on WT DCs, which had emigrated into the culture medium from WT ears, and the CD112 signal in the corresponding KO DC population (Figure 2K–M), confirming the presence of CD112 in migratory DCs. Also, when staining CD112 in dermal DCs that had remained in the cultured ear skin, consistently a higher signal was detected in WT as compared to KO DCs, thus providing further confirmation of the subtle CD112 expression in dermal DCs (Figure 2N–P). Overall, our results showed that CD112 was expressed at high levels in dermal murine LECs. In contrast, dermal DCs and DCs emigrating from murine skin only expressed low levels of CD112.

### 3.3. Loss of CD112 Does Not Impact the In Vivo Migration of Adoptively Transferred or of Endogenous Murine BM-DCs to dLNs

Since we found in vitro transmigration of murine DCs across murine lymphatic endothelium to be reduced in the absence of CD112 (Figure 1), we next investigated DC migration from steady-state or inflamed skin to draining LNs. As an inflammatory model, we elicited a contact hypersensitivity (CHS) response towards oxazolone (4-ethoxymethylene-2-phenyl-2-oxazoline-5–1). Notably, a similar degree of ear swelling and leukocyte infiltration was observed upon inducing a CHS response in the ear skin of WT and CD112 KO mice (Appendix A). Moreover, the weight of the CHS-draining auricular LNs, the LN cellularity, and the numbers of resident DCs (CD11c^+^MHCII^int^) and migratory DCs (CD11c^+^MHCI^hi^) were similar in both genotypes (Appendix A). To further investigate DC migration, a CHS response was next induced in the footpad of WT or CD112 KO mice, followed by adoptive transfer of 1:1 mixtures of WT and CD112 KO BM-DCs, labeled in two fluorescent colors. BM-DC migration to the draining popliteal LNs was analyzed 20 h later by flow cytometry (Figure 3A). When quantifying the ratio of adoptively transferred WT (e.g., Deep red^+^) and CD112 KO (e.g., CMFDA^+^) BM-DCs in popliteal LNs (Figure 3B), we found no difference in this ratio in dLNs, neither upon transfer into steady-state nor into CHS-inflamed footpads of WT or KO mice (Figure 3B,C). Intriguingly, in contrast to what we had observed in CHS-draining auricular LNs (Appendix A), a significant difference in the cellularity of the CHS-draining popliteal LNs between WT and CD112 KO mice was observed, as well as a significant reduction in endogenous LN-resident (CD11c^+^MHCII^int^) and in migratory DCs (CD11c^+^MHCI^hi^) (Appendix A), possibly due to the simultaneous adoptive BM-DC transfer. To further evaluate whether the migration of endogenous DCs from the skin to draining LNs might be compromised, we also performed a FITC painting experiment in TPA-inflamed skin. To this end, TPA was applied to the ear skin of WT and CD112 KO mice, followed by the application of FITC the next day and analysis of the draining auricular LNs 20 h later (Figure 3D). As in the CHS model (Appendix A), the TPA-induced ear swelling response, as well as the weight and cellularity of the ear-draining auricular LNs were similar in WT and CD112 KO mice (Figure 3E–G). Moreover, neither the number of endogenous migratory DCs (CD11c^+^MHCI^hi^) nor the FITC^+^ migratory DCs was affected by the loss of CD112 (Figure 3H–J). Thus, under the experimental conditions tested, no defect in the migration of exogenous BM-DCs or of endogenous FITC^+^ dermal DCs was observed in CD112 ΚO mice.

### 3.4. CD112 Supports Human moDC Transmigration across Lymphatic Endothelium

Previous publications have reported CD112 expression in human DCs [17] as well as in human umbilical vein endothelial cells (HUVECs) [14] and in human LECs [10]. Moreover, treatment with CD112-blocking antibodies was found to reduce in vitro endothelial transmigration of human monocytes, which express the CD112-binding partners CD113 and DNAM-1 [14]. In light of our findings showing a reduction in murine DC transmigration in the absence of CD112 in vitro (Figure 1), we next investigated whether also human in vitro DC transmigration across lymphatic endothelium might be compromised. To this end, human CD14^+^ monocytes were isolated from peripheral blood mononuclear cells (PBMCs) of healthy donors, and DCs were differentiated in vitro using GM-CSF and IL-4. CD14^+^ monocytes and differentiated moDCs were analyzed for expression of CD11c, CD86, CD14 and HLA-DR, and moDCs were found to express higher levels of these markers compared to purified CD14^+^ monocytes (Appendix A). CD112 was present in immature moDCs and further upregulated upon LPS maturation (Figure 4A–C). In contrast, none of the other binding partners of CD112, namely, DNAM-1, TIGIT or CD113, were found to be expressed on immature or LPS-matured moDCs (Figure 4A–C). Similarly, we could only detect expression of CD112, but not of DNAM-1, TIGIT or CD113 on in vitro cultured human LECs (Figure 4D,E). To investigate the contribution of CD112 to human DC transmigration, we performed transwell transmigration experiments with LPS-matured moDCs and human dermal LECs in the presence/absence of a previously described CD112-blocking antibody (R2.525) [10,45]. Notably, human dermal LECs displayed strong CD112 expression, as previously reported [10] (Figure 4D,E). In line with previous findings, the blockade of ICAM-1 significantly reduced human DC transmigration, as evidenced by FACS-based quantification (Figure 4F,G) [28]. Furthermore, we observed that the number of transmigrated DCs was significantly reduced upon CD112 blockade (Figure 4H,I). Thus, similarly to the findings made in the murine system, the blockade of CD112 reduced in vitro transmigration of DC in the human setup.

### 3.5. CD112 Is Expressed by DCs and LECs in Human Skin

Having observed the expression of CD112 on in vitro cultured human moDCs and LECs (Figure 4A–E), we next investigated its expression by endogenous DCs and LECs present in vivo in human skin. To this end, single-cell suspensions were generated from enzymatically digested pieces of human skin obtained from surgical procedures. Specifically, FACS analysis was performed, staining for leukocytes (CD45^+^) as well as for LECs (CD45^−^CD31^+^podoplanin^+^) and BECs (CD45^−^CD31^+^podoplanin^−^) (Figure 5A). This analysis confirmed CD112 expression in both BECs and LECs (Figure 5B). Moreover, we also detected CD112 expression in human dermal DCs, which were identified as HLA-DR^+^CD86^+^ cells (Figure 5C,D). However, similarly as in the murine condition (Figure 2), CD112 expression in DCs seemed lower as compared to its expression in LECs (Figure 5B,D). The presence of CD112 on human DCs was also confirmed by immunofluorescence, when staining frozen human skin sections for CD112 and HLA-DR (Figure 5E). Similarly, CD112^+^ lymphatic vessels and blood vessels could be identified in human skin when co-staining for CD112, the lymphatic marker LYVE-1 and the blood vascular marker PLVAP (Figure 5F).

### 3.6. DC Emigration from Human Skin Is Reduced upon CD112 Blockade

Lastly, to gain further evidence for the contribution of CD112 to the lymphatic migration of human DCs, we performed crawl-out assays [30,38] with human skin. For this, 6 mm punch biopsies were co-cultured for 48 h in a control medium or in the presence of a CD112-blocking antibody (clone R2.525). When analyzing CD112 levels in HLA-DR^+^CD86^+^ DCs that had emigrated from the skin upon culture, a strong CD112 signal was detected by FACS (Figure 5G), indicating that human migratory DCs expressed and possibly even upregulated CD112. Moreover, the number of DCs that had emigrated from punch biopsies cultured in the presence of CD112-blocking antibody was significantly reduced in comparison to the control-treated group (Figure 5H). Interestingly, in addition to the strong expression of CD112, FACS analysis performed on skin-emigrated DCs revealed subtle expression of the CD112-binding partner CD113 in two out of three experiments performed (Figure 5I). Overall, these findings further confirm our hypothesis that CD112 supports human DC migration through dermal afferent lymphatic vessels and suggest the contribution of homophilic CD112 interactions, and potentially also heterophilic CD113–CD112 interactions, to this process.

## 4. Discussion

In this paper, we have investigated the role of CD112 in DC migration across lymphatic endothelium in vitro and in vivo in the murine and human systems. Our results reveal that CD112, which is expressed by murine and human LECs and DCs, supports murine and human DC transmigration across the lymphatic endothelium in vitro and strongly suggests its contribution to the lymphatic migration of DCs in human but not in murine skin.

We have recently found CD112 to be highly expressed by human and murine LECs in vitro as well as by murine lymphatics in vivo [10]. Also, its expression by human in vitro-generated moDCs and human DCs present in LNs has been previously described [14,17]. In contrast to our knowledge, CD112 expression in murine DCs has thus far not been reported, possibly because of its low expression levels in murine DCs. Although CD112 levels were consistently very low in murine BM-DCs, we found that BM-DC in vitro adhesion and transmigration across LN-LECs monolayers were significantly reduced when CD112 expression was lacking on either one or both cell types. Together with the absence of other CD112-binding partners, i.e., of TIGIT, DNAM-1 or CD113, on murine LN-LECs and BM-DCs, these findings suggest that homophilic CD112–CD112 interactions contributed to the observed defect in in vitro transmigration. The fact that we did not see a reduction in the migration of adoptively transferred BM-DCs in vivo likely suggests that in vivo the subtle difference in CD112 expression between WT and KO BM-DCs was compensated by other adhesion molecules contributing to the migratory process. Furthermore, these findings suggest that gene expression of in vitro cultured murine LN-LECs differs from the one of LECs present in dermal lymphatic capillaries in vivo. For example, the latter might express higher levels of other adhesion molecules that can compensate for the loss of CD112 on DCs. Like in the adoptive transfer experiment, we could not detect any evidence of CD112 contributing to the in vivo migration of endogenous dermal DCs in a FITC painting experiment performed in TPA-inflamed ear skin. Notably, as in BM-DCs, CD112 expression was very low on endogenous dermal DCs, supporting our conclusion that the difference between WT and CD112 KO dermal DCs with regards to CD112 expression was too low to have an impact on in vivo migration. Additionally, it is possible that CD112-deficient dermal LECs or DCs upregulate other adhesion molecules to compensate for the loss of CD112. For example, we previously reported that VE-cadherin, a molecule that is key for the junctional integrity of LECs [46,47], is upregulated in dermal LECs of CD112 KO mice [10]. Therefore, constitutive CD112 KO mice might represent a suboptimal model for assessing the contribution of CD112 to migration. However, to our knowledge, no in vivo validated, commercial CD112-blocking antibodies are available that could offer an experimental alternative. Overall, we conclude from these experiments that the in vitro experiments performed with murine BM-DCs and LN-LECs did not faithfully model the in vivo migratory conditions and that CD112 most likely does not contribute to the in vivo migration of murine dermal DCs.

Interestingly, the extent of inflammation elicited by topical application of TPA or by the induction of a CHS response in the ear skin of CD112 KO mice was comparable to the one induced in WT mice. While the application of TPA induces a strong innate inflammatory response, a CHS response primarily depends on the induction of adaptive T-cell immunity. The fact that no difference in the degree of inflammation was observed in both models suggests that neither the induction of T-cell immunity nor recruitment of cells into the inflamed ear skin was compromised in the absence of CD112. Considering that we did not observe any difference in DC migration in CD112 KO mice, these findings might seem intuitive. However, since CD112 also interacts with immune-modulatory molecules like DNAM-1, TIGIT and CD112R [8,14,15,39] and was shown to impact monocyte and T-cell transmigration across blood vascular endothelium [10,14,16], the explanation of the absence of an inflammation phenotype in CD112 KO mice is likely more complex.

Like our findings with murine cells, human DC transmigration across human dermal LEC monolayers was significantly reduced upon CD112 blockade in vitro. However, in comparison to murine BM-DCs and murine dermal DCs, human moDCs as well as DCs present in human skin appeared to express higher CD112 levels. In analogy to our findings in the murine system, we could not detect the expression of any other CD112-binding partner—i.e., DNAM-1, TIGIT or CD113—in cultured human dermal LECs or moDCs. This suggests that also in the human setup, CD112 contributed to in vitro DC transmigration by engaging in homophilic interactions. Since we detected subtle expression of CD113 on skin-emigrated DCs (in two out of three experiments performed), it is possible that in human skin, additionally heterophilic CD113–CD112 interactions contribute to DC migration via lymphatic vessels. Notably, DCs that had emigrated from either murine or human skin in our crawl-out assays expressed higher levels of CD112 in comparison to the large pool of dermal DCs detected by FACS in tissue single-cell suspensions. Considering that such migratory DCs are expected to comprise mainly mature DCs, these findings are in agreement with the CD112 upregulation observed in LPS-matured murine BM-DCs and human moDCs (Figure 1A,B and Figure 4A,B [17]). Importantly, numbers of skin-emigrated human DCs were significantly reduced when human skin punches had been cultured in the presence of a CD112-blocking antibody, in further support of CD112’s role in human dermal DC migration through afferent lymphatics.

Overall, our results indicate that CD112 contributes to human dermal DC migration but not to murine dermal DC migration. The reason for this discrepancy likely lies in species differences in CD112 expression levels between murine and human DCs. Similar to CD112, we recently found that the activated leukocyte cell adhesion molecule (ALCAM), which also mediates homophilic DC–LEC interactions and DC transmigration, is highly expressed in human and murine dermal DCs but only expressed in human but not in murine dermal lymphatics [30]. Although the reasons for such species differences are unknown, the fact that many adhesion molecules seem to collectively support migration could imply that the contribution of one single pathway is by far less critical than, e.g., the contribution of the CCR7–CCL21 axis to DC migration towards lymphatics [23]. Considering that CD112 has many ligands [48], it is probable that LEC-expressed CD112 supports lymphatic migration of other tissue-egressing cell types. Overall, the identification of molecules involved in lymphatic migration could prove beneficial for the modulation of protective immunity, i.e., for enhancing immunity in the context of vaccination or reducing it in the context of organ transplantation.

## Figures and Tables

**Figure 1 cells-13-00424-f001:**
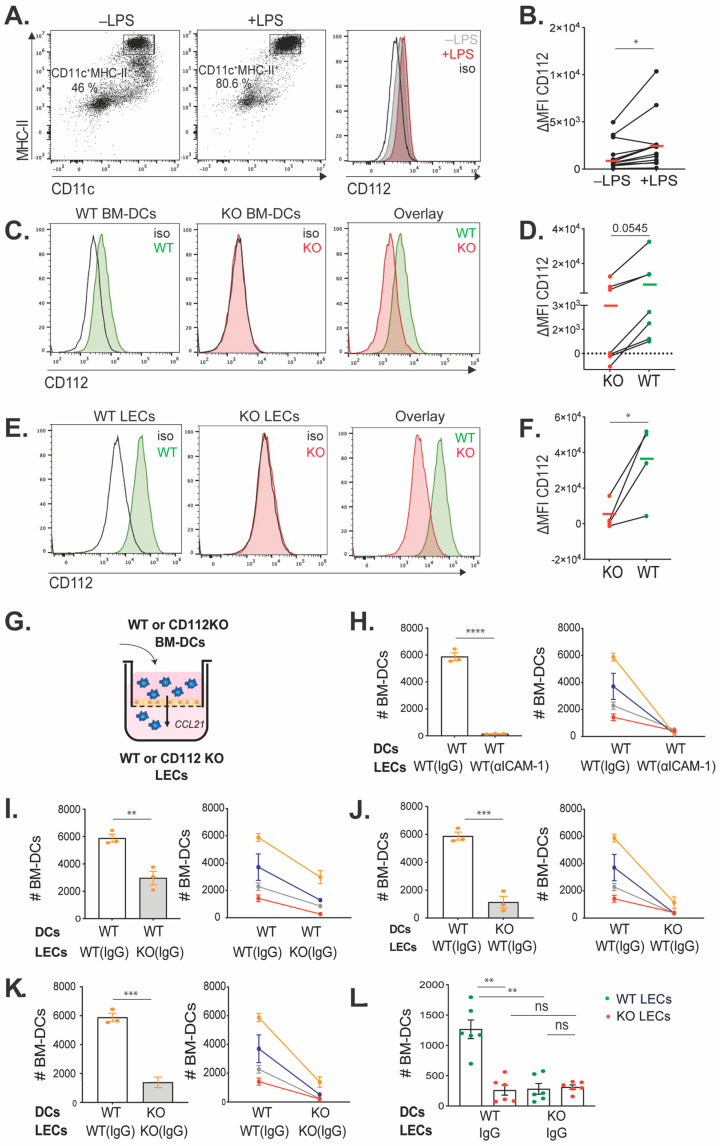
CD112 is expressed in BM-DCs and LECs and supports DC transmigration. (**A**) Flow cytometry analysis of immature (−LPS) and LPS-matured (+LPS) BM-DCs (gated on live/single cells). (**B**) Summary of the delta mean fluorescent intensity (∆MFI; specific-isotype staining) values of CD112 expression of 11 independent experiments. (**C**–**F**) FACS analysis of CD112 expression in (**C**) LPS-matured BM-DCs and (**E**) primary LN-LECs, derived from WT and CD112 KO mice. (**D**,**F**) Summary of the ∆MFI values of CD112 expression of 4–6 independent experiments. Data points of the same experiment in (**B**,**D**,**F**) are connected by a line, and the mean ΔMFI values are indicated by horizontal lines. (**G**) Set up of the transmigration experiments to investigate the transmigration of BM-DCs (WT or KO) across an LEC monolayer (WT or KO). (**H**) Impact of ICAM-1 blockade on transmigration of WT BM-DCs. (**I,J**) Impact of loss of CD112 in either (**I**) LECs or (**J**) BM-DCs on transmigration. (**K**) Impact of simultaneous loss of CD112 in LECs and BM-DCs on transmigration. For each condition in (**H**–**K**), one representative experiment with n = 3 technical replicates is shown on the left, and a summary of the averages of 4 independent experiments (biological replicates, each experiment in a different color) is shown on the right. Data points of the same experiment are connected by a line. (**L**) Adhesion assay of WT and KO BM-DCs to WT or KO lymphatic endothelium. The pool of two independent experiments with three replicates per condition is shown (each dot represents a sample). # BM-DCs: number of BM-DCs. Data in all graphs show mean ± standard error of the mean (SEM). * *p* < 0.05; ** *p* < 0.01; *** *p* < 0.001; **** *p* < 0.0001; ns: not significant.

**Figure 2 cells-13-00424-f002:**
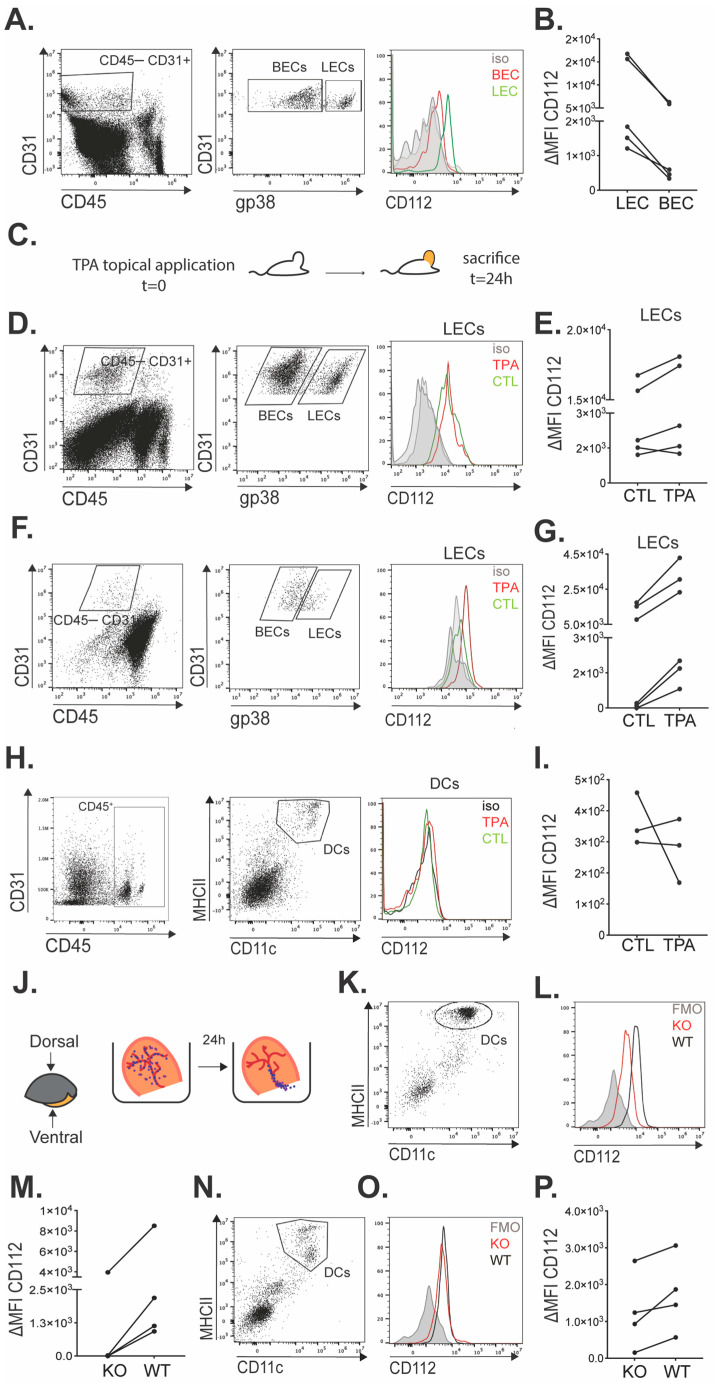
CD112 expression is high in LECs but low in DCs present in murine skin. (**A**,**B**) FACS analysis was performed to detect CD112 expression in dermal LECs and BECs. (**A**) Depiction of the gating strategy in one representative experiment. (**B**) Summary of the delta mean fluorescent intensity (∆MFI; specific-isotype staining) values of CD112 expression observed in 5 independent experiments. (**C**–**G**) Impact of TPA-induced skin inflammation on the expression of CD112 in LECs. (**C**) Schematic depiction of the experiment: Inflammation was induced in the murine ear skin by topical application of TPA and the ear skin and draining auricular LNs analyzed 24 h later. (**D**–**G**) FACS analyses were performed to quantify CD112 expression levels in LECs present in control or inflamed tissues. (**D**,**E**) Analysis of murine ear skin and (**F**,**G**) auricular LN single-cell suspensions. (**E**,**G**) The summary of ∆ MFI values was recorded in 5–6 different experiments performed in one control (CTL) and one TPA-inflamed (TPA) ear skin. (**H**,**I**) FACS gating and quantification of CD112 expression in DCs present in CTL and TPA-inflamed ear skin. (**H**) Gating strategy and (**I**) summary of ∆MFI values recorded in 3 different experiments. (**J**–**P**) Crawl-out experiments. (**J**) Schematic depiction of the experiment performed to evaluate CD112 expression in (**K**–**M**) DCs that had emigrated from murine ear skin into the culture medium or in (**N**–**P**) DCs that had remained in the cultured ear skin at the end of the experiment. Representative (**K**,**N**) FACS dot plots (gating on single/live cells), identifying DCs as MHCII^+^CD11c^+^ cells. (**L**,**O**) Representative histogram plots showing CD112 expression in WT and KO DCs as well as the corresponding fluorescence minus one (FMO) control. (**M**,**P**) Summary of ∆MFI values (defined as specific staining—FMO) recorded in 4 different experiments performed with one WT and one KO mouse each. Data points in (**B**,**E**,**G**,**I**,**M**,**P**) of the same experiment are connected by a line.

**Figure 3 cells-13-00424-f003:**
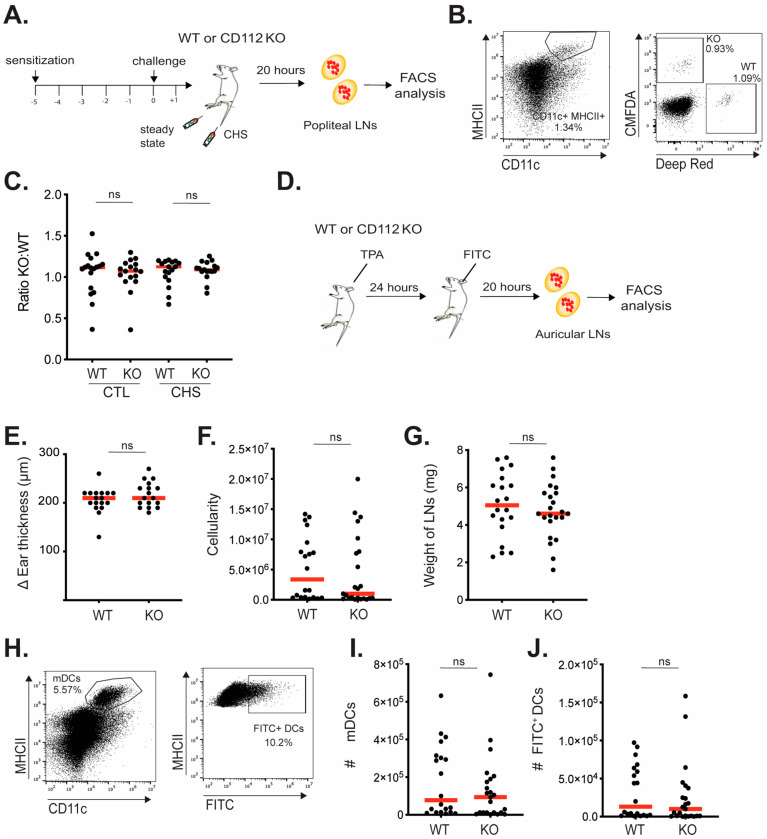
Loss of CD112 does not impact the in vivo migration of adoptively transferred or endogenous DCs to dLNs. (**A**–**D**) Adoptive transfer experiment. (**A**) Scheme of the experiment. (**B**) Gating strategy to identify fluorescently labeled adoptively transferred BM-DCs in popliteal LNs. (**C**) The ratio of KO–WT DCs recovered from popliteal LNs draining control (CTL) or CHS-inflamed (CHS) footpads of WT or KO mice. (**D**–**J**) FITC painting experiment. (**D**) Scheme of the experiment. (**E**) ΔEar thickness, defined as the difference between the ear thickness measured at the start and at the end of the experiment. (**F**) Cellularity and (**G**) weight of the ear-draining auricular LN at the end of the experiment. (**H**) Gating strategy to identify and quantify the number (#) of (**I**) all CD11c^+^MHCII^hi^ migratory DCs (mDCs) and (**J**) FITC^+^ mDCs. Summaries of three (**A**–**D**) and two (**D**–**J**) independent experiments, each with 2–7 mice per condition, are shown. Each dot represents one mouse. Mann–Whitney *t*-test was used. Red bars in all graphs show the mean. ns: not significant.

**Figure 4 cells-13-00424-f004:**
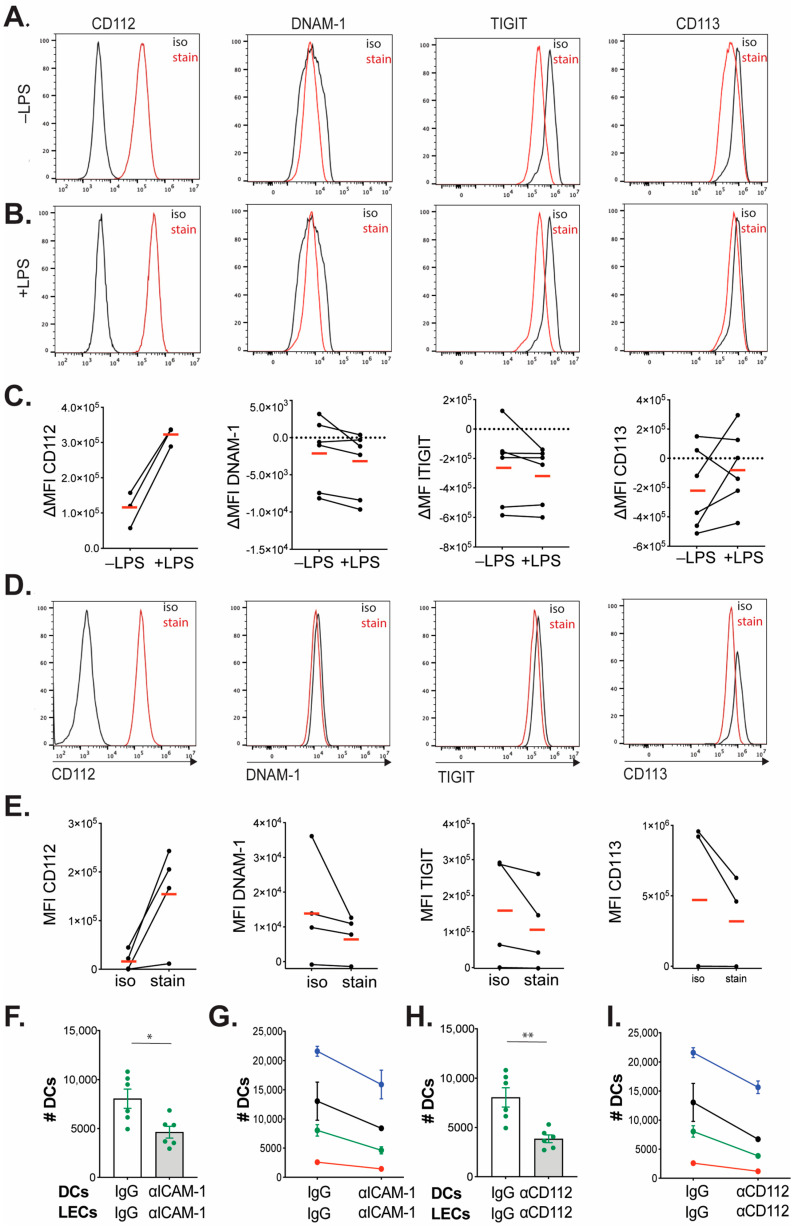
Blockade of CD112 decreases in vitro transmigration of human moDCs across human dermal LEC monolayers. (**A**–**C**) Analysis of CD112, DNAM-1, TIGIT and CD113 expression in in vitro-differentiated (**A**) immature (−LPS) and (**B**) LPS-matured (+LPS) human moDCs. LPS was added 24 h prior to FACS analysis. Representative FACS plots are shown in (**A**,**B**). (**C**) Summary of the delta mean fluorescent intensity (∆MFI; defined as specific-isotype staining) values recorded for each corresponding marker in 3–6 independent experiments (biological replicates). Data points of the same experiment are connected by a line, and the means of the ΔMFI values are indicated by horizontal red lines. (**D**,**E**) Analysis of CD112, DNAM-1, TIGIT and CD113 expression in primary human dermal LECs. (**D**) Representative FACS histograms recorded upon gating on CD31^+^podoplanin^+^ cells, and (**E**) summary of the MFI values recorded for all markers and corresponding isotype controls in 4–5 independent experiments performed on LECs from two different donors. Data points of the same experiment are connected by a line, and the means of the MFI values are indicated by horizontal red lines. (**F**–**I**) Transmigration experiments involving human moDCs and human dermal LECs, performed in the presence/absence of (**F**,**G**) αICAM-1 or of (**H**,**I**) αCD112 or the corresponding isotype controls; (**F**–**I**) The number of transmigrated DCs (# DCs) was assessed. (**F**,**H**) show representative results from one representative experiment with n = 6 technical replicates per condition. (**G**,**I**) show the summaries of four independent experiments (i.e., different biological replicates, shown with different colors) with 3–6 replicates per condition. The averages from each experiment are connected by a line. The standard error of the mean (SEM) is shown; the Mann–Whitney *t*-test was used. * *p* < 0.05; ** *p* < 0.01.

**Figure 5 cells-13-00424-f005:**
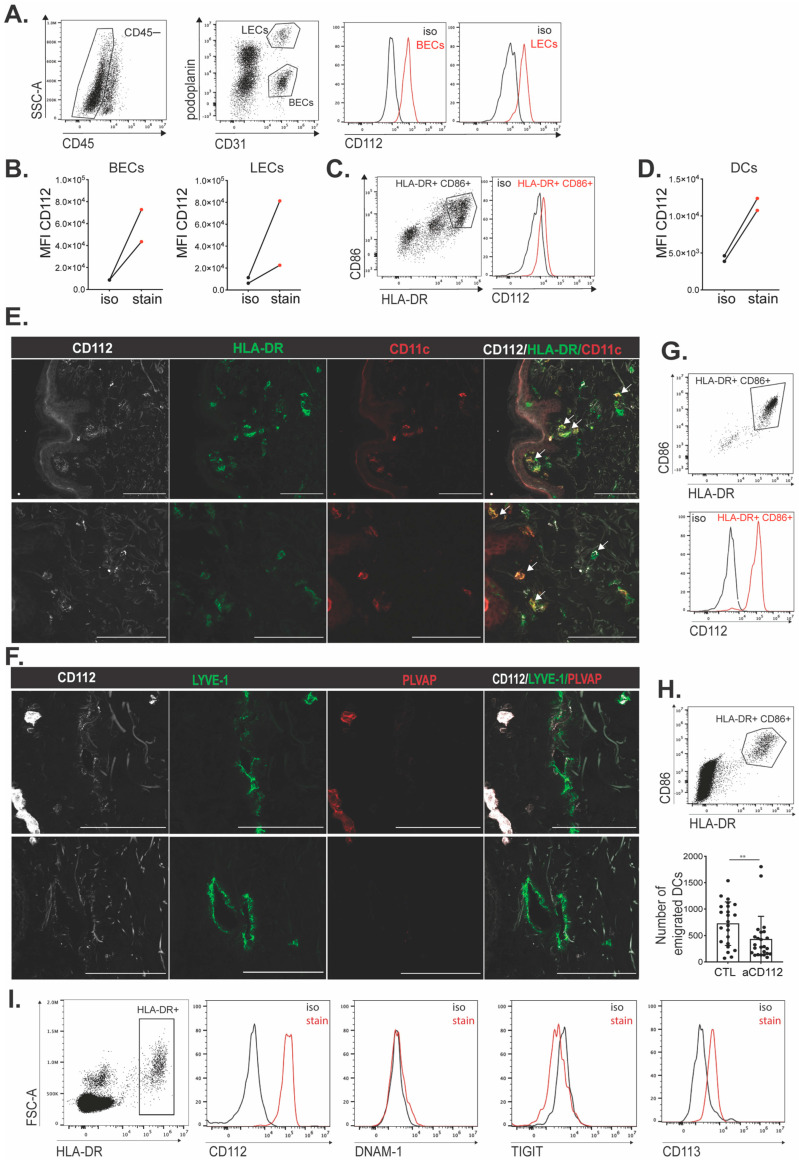
CD112 is expressed by DCs and LECs in human skin. (**A**–**D**) FACS-based analysis of CD112 expression in endothelial cells and DCs present in human skin. (**A**,**C**) Gating strategy used to detect CD112 expression in (**A**) BECs and LECs and (**C**) DCs. (**B**,**D**) Summary of mean fluorescent intensity (MFI) values of CD112 expression in (**B**) LEC and BECs or (**D**) HLA-DR^+^ CD86^+^ DCs in 2 independent experiments (i.e., different biological replicates) was analyzed. Data points of the same experiment are connected by a line. (**E**,**F**) Confocal images of human skin sections depicting (**E**) CD112 expression (white) by dendritic cells (examples indicated by white arrows), identified as HLA-DR^+^ (green) and CD11c^+^ (red). Scale bar = 100 μm (**F**) CD112 expression (white) by lymphatic vessels, LYVE-1 (green) and PLVAP (red). Scale bar = 100 μm. (**G**) Top: Gating strategy and Bottom: representative histogram plot showing CD112 expression on DCs that had emigrated from a human breast skin punch biopsy. (**H**) Crawl-out experiments from punch biopsies derived from either breast or abdominal skin were performed in the presence of a CD112-blocking antibody or media/isotype control (CTL) in the culture medium. Top: Representative FACS gating plot from abdominal skin. Bottom: Quantification of emigrated HLA-DR+CD86^+^ DCs. Pooled data from 5 independent experiments with 4–10 punches per condition are shown. (**I**) Crawl-out experiment from abdominal skin punch biopsies to verify the expression of CD112-binding partners DNAM-1, TIGIT and CD113 on human DCs, identified as live, HLA-DR^+^ cells. Representative stainings from one out of three independent experiments are shown. The mean and standard deviation (SD) are shown in (H). Mann–Whitney *t*-test was used. ** *p* < 0.01.

## Data Availability

Raw data will be made available on the ETH research collection (public repository) after acceptance of the manuscript; https://www.research-collection.ethz.ch.

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
