# Peer review of "CD112 Supports Lymphatic Migration of Human Dermal Dendritic Cells"

_cells, 2024, doi:10.3390/cells13050424_

Round 1

Reviewer 1 Report

Comments and Suggestions for Authors

Trafficking of dendritic cells (DCs) into lymph nodes through lymphatic vessels is essential for appropriate immune responses. The authors previously demonstrated that endothelial cells express CD112 (nectin-2), which plays an important role in angiogenesis and T-cell entry into the spleen. In the present study by Jahromi et al., the authors focused on the role of CD112 in DC trafficking. The authors demonstrated that DCs weakly express CD112 and upregulate the expression upon stimulation. The absence of CD112 on DCs markedly impaired migration in vitro, but such activity was not prominent in skin DC migration. By contrast, the authors demonstrated that the expression of CD112 is significant in human DCs, particularly monocyte-derived DCs, and blockade of CD112 inhibits DC migration through LECs in vitro. Their results mostly support the authors’ claim, and the authors discuss the findings, including the differential role of CD112 on DCs in mice and humans. However, some additional explanations are required to support the findings for the publication.

Major comments:

1.    In the CHS model, although the migration of LPS-stimulated BMDCs generated from CD112 KO mice was not impaired, the authors demonstrated that the number of migratory DCs in the skin dLN was significantly reduced in CD112 KO mice (Fig. 3F). In addition, the number of LN resident DCs were also reduced in CD112 KO mice even without CHS challenge (Fig. 3G). To support the point that CD112 molecule support endogenous DC migration through lymphatics and recruitment of blood-borne DC precursors through blood vessels, the author should additionally show i) the number of DCs in the skin of CD112 KO mice (CTL and CHS), ii) the expression level of CD112 on LN migratory and resident DCs in WT mice (CTL and CHS), iii) CD112 expression on other stromal cell subsets including BEC, HEV, and FRCs, to address if CD112 is implicated in the adhesion of resident or migratory DCs with these stromal cells.

2.    What about the consequence of TPA-induced skin inflammation or CHS induction? If CD112 deficiency affects the amount of DCs through DC migration in the CHS model, some pathological changes could be observed in these models.

3.    CD112 expression were inconsistent in several results. The expression of CD112 in LN was significant in Fig. 1E, while it was marginal in Fig.2F. The difference was also observed between Fig. 2A and 2D. In addition, the authors observed CD112 expression in KO DCs (Fig. 2L, 2O). The authors should improve the results.

4.    In Fig. 4A and supplementary Fig. 2 (upper), did the authors actually characterize PBMCs? Considering the expression pattern of surface markers, they are supposed to be purified CD14+ monocytes.

5.    The authors should also demonstrate the expression of binding partners for CD112 (DNAM-1 and TIGIT) on human DCs and LECs. 

Minor comments

1.    Line 53, please spell JAMs out.

2.    Please rewrite the beginning of the section 3.5. The authors described “We next investigated whether CD112 was also expressed by DCs and LECs in human skin”, but the authors already demonstrated the expression of CD112 on LECs in Fig. 4C.

3.    The explanation of the results in Fig. 4E-F and 4G-H is unclear to the reviewer. What the reviewer understands is that the data in 4E and 4G show representative results from 1 out of 4 independent experiments shown in 4F and 4H, respectively. However, the reviewer could not find the exactly matched results in 4F and 4H (4G and F as well). If the result shown in Fig. 4E was representative of the “green” group in 4F., the average value and the range of error bars must be the same between them. The reviewer suggests checking these results again between authors.

The method for isolating dermal LECs from the ear is missing in the method section. 

Author Response

Reviewer 1

Major comments:

  1. In the CHS model, although the migration of LPS-stimulated BMDCs generated from CD112 KO mice was not impaired, the authors demonstrated that the number of migratory DCs in the skin dLN was significantly reduced in CD112 KO mice (Fig. 3F). In addition, the number of LN resident DCs were also reduced in CD112 KO mice even without CHS challenge (Fig. 3G). To support the point that CD112 molecule support endogenous DC migration through lymphatics and recruitment of blood-borne DC precursors through blood vessels, the author should additionally show i) the number of DCs in the skin of CD112 KO mice (CTL and CHS), ii) the expression level of CD112 on LN migratory and resident DCs in WT mice (CTL and CHS), iii) CD112 expression on other stromal cell subsets including BEC, HEV, and FRCs, to address if CD112 is implicated in the adhesion of resident or migratory DCs with these stromal cells.

Since the data shown in former Fig. 3F,G, which the Reviewer is referring to, had indicated that, in the context of the adoptive BM-DC transfer into CHS-inflamed footpads,  migration of endogenous DCs might be reduced in CD112KO, we performed more experiments to investigate into this topic. Upon inducing a CHS response in the ear skin, no difference in endogenous DC numbers were detected in ear-draining auricular lymph nodes (new data now shown in Supplemental Figure 3A-F). Similarly, performing FITC painting experiments in TPA-inflamed ear skin, we also detected no difference in the numbers of FITC+ or FITC- migratory DCs in draining lymph nodes (new data now shown in revised Figure 3D-J). Since these data further supported our initial conclusion (as already stated in the original version of the abstract), that migration of murine DCs is not compromised in absence of CD112, we suspect that the difference observed in the endogenous DC numbers in the context of the adoptive transfer experiments (former Figure 3F,G) might have been linked with the adoptive transfer. We have therefore decided to move these data to the Supplement (new Supplemental Figure 3G-K) and to rather show the more relevant FITC painting experiments in the main Figure 3.

We have changed the corresponding sections in the Results section (lines 464-492) and also parts of the discussion (lines 618-646) and added the results of the FITC painting experiment to the abstract (line 30). Of note, the expression of CD112 in BECs and HEVs was previously characterized in another study from our group (Russo, Runge et al., Cells 2021). However, in light of the new experimental data revealing no differences in endogenous DC numbers in two inflammation models (Fig. 3D-J and Suppl. Fig. 3A-F), we concluded that this characterization was no longer relevant. With regards to DC numbers in the skin; those were quantified in the context of a CHS-response performed in the ear skin, and no difference was found (Suppl. Fig.2D).

  1. What about the consequence of TPA-induced skin inflammation or CHS induction? If CD112 deficiency affects the amount of DCs through DC migration in the CHS model, some pathological changes could be observed in these models.

As suggested by the Reviewer, we analyzed whether differences in the pathologic response of the two genotypes to TPA- or CHS-induced inflammation might exist. Theses analyses revealed no difference in the ear swelling response or in the dLN weight and cellularity in either model (CHS: Suppl. Figure 2B and Suppl. Figure 3B,C; TPA: Figure 3E-G). In the case of the CHS response, we also quantified leukocyte numbers, including DCs, in the CHS-inflamed ear skin and detected no difference between WT and CD112 KO mice (Suppl. Figure 2).

The new data are now reported in the Results section (line 464-469 and 486-489) and discussed in the Discussion (lines 647-658). 

  1. CD112 expression were inconsistent in several results. The expression of CD112 in LN was significant in Fig. 1E, while it was marginal in Fig.2F. The difference was also observed between Fig. 2A and 2D. In addition, the authors observed CD112 expression in KO DCs (Fig. 2L, 2O). The authors should improve the results.

Fig. 1E vs. 2F: We would like to draw the attention of the Reviewer to the fact that the data shown in Fig. 1E correspond to CD112 expression in in vitro-cultured LN LECs, whereas Fig. 2F shows expression in vivo in LN LECs. Possibly due to the tissue digestion procedure or due to the differences in expression levels between in vitro-cultured and in vivo LN LECs, the expression in vivo in LNs was never as substantial as on the in vitro cultured LN LECs. By contrast, at the level of the dermal lymphatics, where entry of DCs into afferent lymphatics is supposed to happen (i.e. the process we are simulating with our in vitro transmigration experiments), LEC expression of CD112 was very clear and strong (Fig. 2E).

Fig. 2A vs. 2D: In case of dermal LECs (Figure 2A, 2D), we show and report that CD112 is consistently expressed in steady-state (2A – green histogram line and corresponding quantification), what is also seen in the steady-state data presented in Fig. 2D (green histogram line and corresponding quantification). These data are consistent, in our opinion.

Fig. 2L vs. 2O: We thank the Reviewer for noticing this mistake. Since we were comparing WT to KO in Fig. 2L and 2O, we did not use an isotype in these experiments but rather compared to the unstained signal in the fluorescence channel used for detecting CD112 (i.e. fluorescence minus one, FMO). FMO signals are typically lower than signals obtained when using isotype antibodies. Therefore, even if there is a difference between the FMO and the CD112 staining in KO DCs, this cannot be interpreted as CD112 being present on KO. Rather, if there is a difference between WT and KO (as seen in Fig. 2L), this indicates CD112 expression in WT DCs.  We have now corrected the Figure and also the legend text, mentioning FMO instead of isotype (Lines 454-455).  

  1. In Fig. 4A and supplementary Fig. 2 (upper), did the authors actually characterize PBMCs? Considering the expression pattern of surface markers, they are supposed to be purified CD14+ monocytes.

We thank the Reviewer for also spotting this mistake. The FACS plots originally shown in Fig. 4A and former Suppl. Fig. 2 (now Suppl. Fig. 4) indeed did not show PBMCs but corresponded to the CD14+ monocytes that were isolated from PBMCs. In the case of the Suppl. Figure (now Suppl. Fig. 4) the label and Figure legend have now been corrected. In the case of Fig. 4, we decided to take out the monocyte data (now mentioned as “data not shown” – line 519), to make space for the FACS plots and quantifications of expression of other CD112 binding partners in human DCs and LECs, as requested by the Reviewer in the next point.

  1. The authors should also demonstrate the expression of binding partners for CD112 (DNAM-1 and TIGIT) on human DCs and LECs.

Sparked by the comment of the Reviewer we have additionally stained in vitro cultured human monocyte-derived DCs and LECs for DNAM-1, TIGIT and also CD113 (see revised Fig. 4A-E). These experiments did not detect expression of any other CD112 binding partner in human LECs and also not in immature or LPS-matured human monocyte-derived DCs. Please note that these findings are identical to what we had observed for murine BM-DCs and primary LN LECs (Supplemental Figure 1).

To gain even more insights, we now also stained for DNAM-1, TIGIT and CD113 on endogenous human dermal DCs that had emigrated from human skin into culture medium (see revised Fig.5I). Interestingly, these experiments revealed a subtle expression of CD113 on emigrated DCs in 2 out of 3 experiments performed (Fig. 5I). This suggests that, in addition to homophilic CD112-CD112 interactions, also CD112-CD113 interactions likely support DC migration via lymphatic vessels in vivo. We have now integrated these new findings into the Results section (lines 521 – 524 and 580-586) and Discussion (lines 662-668).

Minor comments

  1. Line 53, please spell JAMs out.

We have now spelled out JAMs (junctional adhesion molecules.)

  1. Please rewrite the beginning of the section 3.5. The authors described “We next investigated whether CD112 was also expressed by DCs and LECs in human skin”, but the authors already demonstrated the expression of CD112 on LECs in Fig. 4C.

We thank the Reviewer for this comment. While we demonstrated expression of CD112 on in vitro cultured dermal LECs in Fig. 4C, we had not yet done so on cells in the tissue context, in vivo. The first sentence of the section 3.5 has now been re-written to make this clearer (line 556-558).

  1. The explanation of the results in Fig. 4E-F and 4G-H is unclear to the reviewer. What the reviewer understands is that the data in 4E and 4G show representative results from 1 out of 4 independent experiments shown in 4F and 4H, respectively. However, the reviewer could not find the exactly matched results in 4F and 4H (4G and F as well). If the result shown in Fig. 4E was representative of the “green” group in 4F., the average value and the range of error bars must be the same between them. The reviewer suggests checking these results again between authors.

We thank the Reviewer for spotting the discrepancy in the data! – The reason why the error bars in the “green” datasets did not match between Figure 4K,L and 4M,N is because – by mistake – the standard deviation (SD) and not the standard error of the mean (SEM) had been plotted in Figures 4K and 4M. This has now been corrected, i.e. SEMs are shown in all plots, as indicated in the Figure legend (line 553).   

The method for isolating dermal LECs from the ear is missing in the method section.

Since we do not have a robust protocol established for isolating murine LECs from skin, we performed all experiments with primary LECs isolated from LNs. The corresponding protocol is described in the methods section in lines 131-144.

Reviewer 2 Report

Comments and Suggestions for Authors

The manuscript from Neda Haghayegh Jahromi et al. investigated the expression of CD112 in mouse and human LEC and BM derived and monocytes-derived DCs and its in vitro and in vivo contribution Dc migration. They found that CD112 was highly expressed in murine dermal LECs and that CD112 levels were low on endogenous dermal DCs and BM-DCs. They also found that transmigration of BM-DCs across or adhesion to murine LEC monolayers was reduced when CD112 was absent on LECs, DCs or both cell types. Similarly, the in vitro transmigration of human DCs across human LECs was significantly reduced upon CD112 blockade. These in vitro data were supported using in vivo experimental approaches. Overall, these data demonstrated the contribution of CD112 to DC migration.

Major concerns: 

Bone marrow derived DC were generated with GM-CSF alone however monocytes derived DC were generated with GM-CSF and IL-4. Recent evidence however, has highlighted that BMDCs generated with GM-CSF are highly heterogeneous and contain cells that do not resemble any known in vivo DC subset. A substantial proportion of these cells even appear closer to macrophages than DCs. Why the authors did not use IL-4 and GM-CSF to generate mouse  DCs as did fir human DC. Could some mouse data be explained by the fact that not all mouse DCs are not pure DCs.

Line 330 (Fig1A, B). Please provide the mean of MFI and statistics as well. Same comment applies to Figs 1C and 1E. 

line 332 (Fig 1C, D): Are BM.DCs stimulated or not in these experiments?

Not clear why the authors showed immature and mature DCs profiles but not the CD112 expression in both conditions.

 Line 355: There is no Supplementary Figure S1A. (there is a Sup Fig.1). Please add legends to Sup Fig. 1 and Sup Fig.2. Again, Are BM.DCs simulated or not in these experiments?

Not clear why the authors are showing the expression of CD113 on imELCs which does not support this author's conclusion reached based on murine BM.DCs and LECs. 

Line 389 (Fig. 2D, E) The increase in CD112 level as showed in FACS plots (fig 2D) is not convincing.

 Line 392 (Fig. 2H, I). How the authors explain the clear shift in CD112 expression in FACS panel of Fig.2 H. The shift in this fig is more evident than in Fig.2 D. How the authors explain this discrepancy?

Line 401 (Fig. 2K-M). Do WT mice develop similar or more ear inflammation than CD112 KO mice?

Is there less number migratory CD112 KO DC in culture medium than WT DCs?

Line 402. Did the authors checked other the migratory DC markers?

 Line 405 (Fig.2N-P). CD112 KO mice are not supposed to express CD112 or if they do they should express much less than WT mice.

Line 446 (Fig 3D) If the cellularity is reduced in the CSH-draining PLNs in CD112 KO mice, is the inflammation less or more severe in these mice?

Line 449-450 (Fig 3F,G) Did the authors checked another marker that distinguish migratory and resident DCs?

Line 470.  How the authors reconcile in vitro data where they show a reduction of DC transmigration in absence of CD112 (Fig.1) but not in vivo (Fig. 3) ?

 Line 525 (fig 5 I).  The strong CD112 signal detected in human DCs was not seen in mouse DCs (where CD112 levels is lower than in LEC). Is it possible that the method how murine BM.DCs were generated (with GM-CSF alone) explain this difference in CD112 expression levels? I would be interesting to investigate whether CD112 level in BMDCs generated with GM-CSF and IL-4?

 Line 623-624. (Supplementary Materials) no title, no table and no video are shown or described in the current manuscript. Please correct accordingly.

Author Response

Reviewer 2

Major concerns:

Bone marrow derived DC were generated with GM-CSF alone however monocytes derived DC were generated with GM-CSF and IL-4. Recent evidence however, has highlighted that BMDCs generated with GM-CSF are highly heterogeneous and contain cells that do not resemble any known in vivo DC subset. A substantial proportion of these cells even appear closer to macrophages than DCs. Why the authors did not use IL-4 and GM-CSF to generate mouse DCs as did fir human DC. Could some mouse data be explained by the fact that not all mouse DCs are not pure DCs.

We thank the Reviewer for this valuable comment. It is well possible that, depending on the protocol used, it might have been possible to generate DCs in vitro with higher CD112 expression levels. However, considering that also endogenous murine DCs expressed extremely low CD112 levels and that we found no strong in vivo evidence (see also the experiments performed to response to the questions of Reviewer 1) for the contribution of CD112 to DC migration in vivo, we decided to not investigate into this issue.

Line 330 (Fig1A, B). Please provide the mean of ΔMFI and statistics as well. Same comment applies to Figs 1C and 1E.

In response to the Reviewer’s comment, we have performed additional experiments to reach statistical significance for the data shown in Figures 1A,B and 1E,F and a near-significance (p=0.0545) in the case of Fig. 1C,D (please note that n=8 experiments with the same outcome were performed in this case). We are now also indicating the mean of the ΔMFIs in the graphs.

Line 332 (Fig 1C, D): Are BM.DCs stimulated or not in these experiments?

The data shown derived from LPS-matured BM-DCs. The information has now been added to the Results section (Line 355) and Figure Legend (Line 389). We thank the Reviewer for noticing this oversight.

Not clear why the authors showed immature and mature DCs profiles but not the CD112 expression in both conditions.

The expression of CD112 in immature and mature BM-DCs is shown in Fig. 1A,B.

Line 355: There is no Supplementary Figure S1A. (there is a Sup Fig.1).

This has been corrected.

Please add legends to Sup Fig. 1 and Sup Fig.2. Again, Are BM- DCs simulated or not in these experiments?

We apologize for this oversight. The legends have now been added. The BM-DCs shown in the analysis in Suppl. Fig. 1 were LPS-matured.

Not clear why the authors are showing the expression of CD113 on imLECs which does not support this author's conclusion reached based on murine BM. DCs and LECs.

The staining of CD113 on imLECs (immortalized LECs) was used as a positive control, to demonstrate that the used anti-CD113 antibody worked. In the transmigration assays the primary LECs isolated from LNs were used, which did not express CD113 (Suppl. Fig. 1). This information had been written in the Supplemental Figure legends, which unfortunately had not been uploaded together with the Supplemental Figures. This has now been corrected. Notably, also primary human dermal LECs did not express CD113 (see revised Fig. 4.D,E).

Thus, in our opinion, the conclusion of our in vitro experiments, i.e. that BM-DC transmigration across primary LN LECs (Fig. 1G-L) most likely involves homophilic CD112 interactions, remains valid.

Line 389 (Fig. 2D, E) The increase in CD112 level as showed in FACS plots (fig 2D) is not convincing.

In response to the Reviewer’s comment we have revised the text in the results section and now state that CD112 was only slightly upregulated in 4 out of 5 experiments (Lines 411-414.)

Line 392 (Fig. 2H, I). How the authors explain the clear shift in CD112 expression in FACS panel of Fig.2 H. The shift in this fig is more evident than in Fig.2 D. How the authors explain this discrepancy?

In Fig. 2H, the one experiment that showed a subtle upregulation had been shown in the Figure panel. But as can be seen from Fig. 2I, this was not consistent. We are now showing one of the other experiments, to not provide a false impression that CD112 might have been upregulated in DCs by TPA.

Line 401 (Fig. 2K-M). Do WT mice develop similar or more ear inflammation than CD112 KO mice?

The experiment shown in Figure 2K-M was performed in absence of experimentally induced inflammation. However, sparked by the comment of the Reviewer, we addressed this question by comparing TPA- or CHS-induced ear skin inflammation in WT and CD112 KO mice. Theses analyses revealed no difference in the ear swelling response or in the dLN weight and cellularity in either model (CHS: Suppl. Figure 2B and Suppl. Figure 3B,C; TPA: Figure 3E-G). In the case of the CHS response, we also quantified leukocyte numbers, including DCs, in the CHS-inflamed ear skin and detected no difference between WT and CD112 KO mice (Suppl. Figure 2).

The new data are now reported in the Results section (line 464-469 and 486-489) and discussed in the Discussion (lines 647-658). 

Is there less number migratory CD112 KO DC in culture medium than WT DCs?

In line with no evidence for murine DC migration in vivo (Figure 3 and Suppl. Figure 3), we saw no difference in DC numbers in the culture medium. The data is now mentioned as “data not shown” (Lines 429-430).

Line 402. Did the authors checked other the migratory DC markers?

No, we did not check into any further migratory sub-populations

Line 405 (Fig.2N-P). CD112 KO mice are not supposed to express CD112 or if they do they should express much less than WT mice.

We thank the Reviewer for reading the manuscript so carefully and spotting this mistake (in fact, both Reviewers did)! – Since we were comparing staining intensities of WT to KO in experiments shown in Fig. 2J-P, we did not use an isotype in these experiments but rather compared to the unstained signal in the CD112 channel (i.e. fluorescence minus one, FMO). FMO signals are typically lower than signals obtained when using isotype antibodies. Therefore, even if there is a difference between the FMO and the CD112 staining in KO DCs, this cannot be interpreted as CD112 being present on KO. Rather, if there is a difference between WT and KO (as seen in Fig. 2L), this indicates CD112 expression in WT DCs. 

We have now corrected this information in the representative FACS panels (Fig. 2L & Fig. 2O) and also in the Figure Legend text (lines 454-455).

Line 446 (Fig 3D) If the cellularity is reduced in the CSH-draining PLNs in CD112 KO mice, is the inflammation less or more severe in these mice?

In response to the questions of both Reviewers, we investigated the inflammatory response by comparing the extent of TPA- or CHS-induced ear skin inflammation in WT and CD112 KO mice. These experiments revealed no difference in the degree of ear swelling, leukocyte infiltration (measured in the case of CHS) or the weight and cellularity of the ear-draining popliteal lymph node. These results are now displayed in Suppl. Fig. 2 & Suppl. Fig. 3A-C (CHS response) and Fig. 3D-G (TPA model), reported in the Results section (line 464-469 and 486-489) and in the Discussion (lines 647-658).

Line 449-450 (Fig 3F,G) Did the authors checked another marker that distinguish migratory and resident DCs?

No, we only checked MHCII and CD11c since these markers are conventionally used to differentiate the LN-resident and migratory DCs (see also Ohl et al., Immunity 2004 or Eisenbarth, Nat. Rev. Immunology 2019).

Line 470.  How the authors reconcile in vitro data where they show a reduction of DC transmigration in absence of CD112 (Fig.1) but not in vivo (Fig. 3)?

Sparked by the comment of the Reviewer, we have re-written most of the Discussion, to make our interpretation of the data clearer (lines 609-691). Moreover, we have also changed two sentences in the Abstract for clarity (lines 30, 31 and 36).

 Line 525 (fig 5 I).  The strong CD112 signal detected in human DCs was not seen in mouse DCs (where CD112 levels is lower than in LEC). Is it possible that the method how murine BM.DCs were generated (with GM-CSF alone) explain this difference in CD112 expression levels? I would be interesting to investigate whether CD112 level in BMDCs generated with GM-CSF and IL-4?

Since endogenous murine dermal DCs also expressed extremely low levels of CD112, and since we did not observe and evidence for an in vivo migration defect in the murine system, we concluded that it makes little sense to further investigate how different BM-DC generation protocols would impact CD112 expression levels in vitro.

Conversely, in the case of the human DCs, CD112 expression levels were higher both in in vitro-differentiated monocyte-derived DCs and in endogenous dermal DCs. We there conclude that CD112 most likely plays no role in murine DC migration, likely due to species differences in the level of CD112 expressed by DCs (see Discussion line 689-691).

Line 623-624. (Supplementary Materials) no title, no table and no video are shown or described in the current manuscript. Please correct accordingly.

We apologies for this mistake; unfortunately, the Supplemental Figures had been uploaded without the corresponding Figure legends. This has now been corrected.

Round 2

Reviewer 1 Report

Comments and Suggestions for Authors

The authors clearly addressed my concerns and improved their story. I have some minor comments on the revised manuscript.

1.    Supplementary figure 3 was missing. Please correct it.

2.    The authors would be better to avoid frequent usage of “No difference ….. was found (particularly in the section 3.3)”.

Comments on the Quality of English Language

See above comment #2. 

Author Response:

1.    Supplementary figure 3 was missing. Please correct it.

Here is the file of the Supplementary figures, apologies, must have been a copy-paste issue.

2. Please carefully check Reviewers proposal: “The authors would be better to avoid frequent usage of “No differ-ence ….. was found (particularly in the section 3.3)”.

Response: We revised 3 out of 4 sentences in this paragraph to avoid the usage of “no difference”.